# Nitric Oxide Mediation in Hydroxyurea and Nitric Oxide Metabolites’ Inhibition of Erythroid Progenitor Growth

**DOI:** 10.3390/biom11111562

**Published:** 2021-10-21

**Authors:** Tijana Subotički, Olivera Mitrović Ajtić, Dragoslava Djikić, Marijana Kovačić, Juan F. Santibanez, Milica Tošić, Vladan P. Čokić

**Affiliations:** 1Department of Molecular Oncology, Institute for Medical Research, National Institute of Republic of Serbia, University of Belgrade, 11129 Belgrade, Serbia; tijana@imi.bg.ac.rs (T.S.); oliveram@imi.bg.ac.rs (O.M.A.); dragoslava@imi.bg.ac.rs (D.D.); marijana.buac@imi.bg.ac.rs (M.K.); jfsantibanez@imi.bg.ac.rs (J.F.S.); milica.tosic@imi.bg.ac.rs (M.T.); 2Centro Integrativo de Biología y Química Aplicada, Universidad Bernardo O’Higgins, Santiago 8370993, Chile

**Keywords:** hydroxyurea, nitric oxide, erythroid colonies, nitric oxide synthase, nitrite, nitrate

## Abstract

In several systems, hydroxyurea has been shown to trigger nitric oxide (NO) release or activation of NO synthase (NOS). To elucidate this duality in its pharmacological effects, during myelosuppression, we individually examined hydroxyurea’s (NO releasing agent) and NO metabolites’ (stable NO degradation products) effects on erythroid colony growth and NOS/NO levels in mice using NO scavenger 2-phenyl-4,4,5,5-tetramethylimidazoline-1-oxyl-3-oxide (PTIO). Hydroxyurea and nitrite/nitrate decreased the bone marrow cellularity that was blocked by PTIO only for the NO metabolites. Hydroxyurea inhibition of colony-forming unit-erythroid (CFU-E) formation and reticulocytes was reversed by PTIO. Moreover, hydroxyurea, through a negative feedback mechanism, reduced inducible NOS (iNOS) expressing cells in CFU-E, also prevented by PTIO. Nitrate inhibition of burst-forming units-erythroid (BFU-E) colony growth was blocked by PTIO, but not in mature CFU-E. The presented results reveal that NO release and/or production mediates the hydroxyurea inhibition of mature erythroid colony growth and the frequency of iNOS immunoreactive CFU-E.

## 1. Introduction

Nitric oxide (NO) is an intracellular metabolite of hydroxyurea used as a therapeutic in sickle cell anemia [1,2]. In preclinical *in vitro* studies, hydroxyurea increases endothelial cell production of NO by stimulating endothelial NO synthase (eNOS) [3], while the later study showed that hydroxyurea enhances nitrite production only in combination with heme [4]. In addition, hydroxyurea regulation of erythroid cell proliferation and apoptosis is NOS dependent [5]. Hydroxyurea *in vitro* decreases the growth of erythroid colonies with the participation of NOS [6,7]. NO donors also inhibit the growth of erythroid and myeloid colonies derived from bone marrow mononuclear and CD34^+^ cells [8,9]. In preclinical animal studies, hydroxyurea reduction in leukocyte adhesion and extravasation is reversed by the NO scavenger, but not by NOS inhibition [10].

In clinical studies, as the stable degradation products of NO, plasma levels of nitrite and nitrate are increased in patients with sickle cell disease and essential thrombocythemia on receiving hydroxyurea therapy [1,11]. The production of total NO levels (NOx) by hydroxyurea is supported by the NOS substrate l-arginine in blood [12]. In contrast, hydroxyurea can enhance the NO plasma levels by a mechanism that does not involve erythrocyte NOS activity [13].

NO is rapidly (milliseconds) oxidized to the stable-end metabolites, nitrite and nitrate, in biologic systems [14]. The half-lives of nitrate and nitrite in the circulation are about 5–8 h and 20–45 min, respectively [15]. As the final products of NO oxidation pathways and reversible NO metabolites [14], nitrite and nitrate may have the potential to distract or overlap hydroxyurea’s efficacy as a NO releasing agent.

We now question whether hydroxyurea primarily acts as an NO donor or cumulatively increases the reversible NO metabolites participating in myelosuppression. To observe this, we use mice for *in vivo* parallel studies with hydroxyurea and the final products of NO (nitrite and nitrate) in co-treatment with a NO chelator PTIO that does not influence the NOS activity. We examine the effects of the NO dependence in hydroxyurea on the myeloid progenitor growth and peripheral blood cells. The mouse bone marrow cells are used for ex vivo expansion of myeloid cultures to analyze the impact of hydroxyurea and NO metabolites on the frequency of NOS isoforms.

## 2. Materials and Methods

### 2.1. Colony Forming Assays

The experimental protocol was already described [5] and approved by the Ethics Committee of the Institute for Medical Research, University of Belgrade, Serbia, according to the National Law on Animal Welfare, consistent with guidelines for animal research and principles of the European Convention for the Protection of Vertebrate Animals Used for Experimental and Other Purposes (Official Daily No. 323-07-02569/2018-05). Briefly, the male CBA mice (6–8 weeks old) were treated daily, via the tail vein, with NO scavenger 2-phenyl-4,4,5,5-tetramethylimidazoline-1-oxyl 3-oxide (PTIO, 1 mg/kg, Santa Cruz Biotechnology, Dallas, TX, USA, in Aqua Pro Injectione (Institute of Immunology and Virology “TORLAK”, Belgrade, Serbia)) and hydroxyurea (200 mg/kg in phosphate-buffered saline (PBS), Sigma-Aldrich, St. Louis, MI, USA) with or without preincubation of 30 min with PTIO (1 mg/kg, Santa Cruz Biotechnology) for seven consecutive days. In addition, animals were treated with sodium nitrite (1 mmol/kg in PBS, Sigma-Aldrich) or sodium nitrate (1 mmol/kg in PBS, Sigma-Aldrich) with or without 30 min preincubation with PTIO (1 mg/kg, Santa Cruz Biotechnology). The total volume injected on each mouse was 10 µL/g (for one shot, per 5 µL/g for two shots). Then, 6 × 10^4^/mL bone marrow cells were plated in methylcellulose media (StemCell Technologies, Vancouver, BC, Canada) containing 3 U/mL erythropoietin (EPO, StemCell Technologies, MethoCult M3334), and 3 × 10^4^/mL cells were plated in methylcellulose media containing 3 U/mL EPO supplemented with 50 ng/mL SCF, 10 ng/mL IL-3, and 10 ng/mL IL-6 (StemCell Technologies, MethoCult GF M3434). Following an incubation period of seven days in MethoCult GF M3434 medium, burst-forming unit-erythroid (BFU-E) and colony-forming unit-granulocyte/macrophage (CFU-GM) colonies were enumerated using an inverted microscope (Olympus CKX41, Tokyo, Japan). Colony-forming unit-erythroid (CFU-E) colonies were scored after three days of culture in MethoCult M3334.

### 2.2. Hematologic Parameters

Mouse blood was collected from the retro-orbital sinus. Mouse blood cell counts were made using a hemocytometer, as previously described [5]. Hemoglobin was analyzed with the cyanmethemoglobin method and determined spectrophotometrically (RT-6100 Microplate Reader, Rayto, Shenzhen, China), while hematocrit was calculated after brief centrifugation.

### 2.3. Measurement of Nitrite Levels

Plasma nitrite (NO_2_-) and nitrate (NO_3_) levels were measured by the Griess method, as previously described [5]. Briefly, 100 µL of each sample and each standard were pipetted into separate wells. Fifty microliters of 2% sulfanilamide (in 5% HCl) were added, followed by 50 µL of 0.1% N-(1 naphthyl) ethylenediamine dihydrochloride (NEDD) in deionized water. After incubation, the absorbance of the reaction mix was read at 540 nm by spectrophotometer (RT-6100 Microplate Reader, Rayto).

### 2.4. Immunocytochemistry

Mononuclear cells were separated from the bone marrow cells with lymphocyte separation medium (LSM, Capricorn Scientific, Ebsdorfergrund, Germany) according to the manufacturer’s instruction. For cytoplasmatic staining, mice bone marrow cells as well as CFU-E and BFU-E / CFU-GM colonies from methylcellulose cultures were collected onto microscope glass slides, as previously described [5]. Briefly, after fixation and incubation with the anti-neuronal NOS (nNOS, cat no. sc-5302 (1:100), Santa Cruz Biotechnology), anti-inducible NOS (iNOS, cat. no. sc-651 (1:100), Santa Cruz Biotechnology), and anti-eNOS antibodies (cat. no. sc-654 (1:100), Santa Cruz Biotechnology) immunostaining was performed using a computer-supported imaging system (analysis Pro 3.1) connected to the light microscope (Olympus AX70, Hamburg, Germany) with an objective magnification of ×40.

### 2.5. Statistical Analysis

The one-way ANOVA and Dunnett’s post test were applied using Prism 6 software (GraphPad Software Inc., San Diego, CA, USA). The results are expressed as the mean ± SEM, and differences at *p* < 0.05 are accepted as the level of significance.

## 3. Results

### 3.1. Plasma NO Derivative Levels Following In Vivo Treatment of Mice by Hydroxyurea

To examine the long-term effect on NO levels in mice, we measured the NO metabolite levels in the peripheral blood plasma of mice after seven days of treatment with hydroxyurea. Daily treatment of hydroxyurea did not change NO_3_ and NOx (nitrite and nitrate) levels, while the NO scavenger PTIO did not significantly reduce NO_3_- and NOx levels in co-treatment with hydroxyurea (Figure 1A). NO scavenger PTIO significantly reduced the NO_3_ and NOx levels, but not in co-treatment with nitrate (Figure 1B,C). Reduced plasma NO_2_ levels, by PTIO, were increased in the presence of hydroxyurea but did not reach statistical significance (not shown). Overall, hydroxyurea did not induce plasma NO metabolites during prolonged *in vivo* treatment of mice.

### 3.2. Hydroxyurea and NO Effects on Bone Marrow and Peripheral Blood Cells of Mice

To explore the cytostatic and NO-dependent properties of hydroxyurea in the hematopoietic microenvironment and circulation, bone marrow was examined in mice treated with hydroxyurea for seven days. Hydroxyurea demonstrated a significant decrease in bone marrow cellularity which was not prevented by PTIO (Figure 2A). Similarly, both sodium nitrite and sodium nitrate decreased the bone marrow cellularity that was blocked by PTIO (*p* < 0.01, Figure 2B). PTIO cut by half the hydroxyurea’s reduction in reticulocytes (*p* < 0.01, Table 1). PTIO prevented the increase in reticulocytes caused by sodium nitrite and sodium nitrate (Table 1). The cytostatic effects of hydroxyurea were confirmed in both bone marrow and peripheral blood, regulated by NO levels for reticulocytes in peripheral blood.

### 3.3. Effects of Hydroxyurea and NO Metabolites on the Myeloid Progenitor Growth

The cytotoxic and NO-dependent effects of hydroxyurea were examined during the myeloid progenitor growth and differentiation. Bone marrow cells harvested from mice, treated with hydroxyurea for seven days, showed a reduced quantity of immature BFU-E colonies, but chelator PTIO failed to protect them (Figure 3A). Nitrite and nitrate inhibition of the immature BFU-E colony growth was diminished by PTIO (Figure 3B). Hydroxyurea demonstrated a fivefold reduction in the number of CFU-E colonies completely reversed by PTIO (*p* < 0.001, Figure 3C). The applied concentration of nitrite and nitrate also reduced the quantity of mature CFU-E colonies not blocked by PTIO (Figure 3D). Hydroxyurea inhibited the growth of CFU-GM colonies, which was not affected by PTIO (*p* < 0.01, Figure 3E). In addition, NO metabolites also inhibited the CFU-GM colony growth (*p* < 0.001), reversed by PTIO only for nitrates (*p* < 0.05, Figure 3F). Hydroxyurea and NO metabolites generally inhibited the myeloid colony growth, indicating NO dependence for hydroxyurea in the CFU-E colonies and nitrate (NO_3_) in the BFU-E and CFU-GM colonies in applied concentrations.

### 3.4. The Level of NOS Isoforms in Bone Marrow Cells during Hydroxyurea and NO Metabolites Treatment

To observe the impact of hydroxyurea and NO metabolites on NOS enzymes in the hematopoietic microenvironment, we analyzed NOS isoforms in bone marrow cells during prolonged treatment of mice with hydroxyurea and sodium nitrite/nitrate. The frequency of immunoreactive eNOS cells was reduced by hydroxyurea (*p* < 0.01) and nitrite/nitrate (*p* < 0.001, Figure 4A,B). The reduction in eNOS positive cells was partially reversed by PTIO during sodium nitrite (NO_2_) treatment (*p* < 0.05, Figure 4B). The frequency of nNOS expressing cells was also significantly reduced by hydroxyurea (*p* < 0.05, Figure 4C) and nitrite (*p* < 0.01, Figure 4D), and prevented by PTIO in nitrite treatment (*p* < 0.01, Figure 4D). The NO scavenger PTIO decreased iNOS positive cells during hydroxyurea treatment (*p* < 0.05, Figure 4E). Sodium nitrite/nitrate decreased the frequency of iNOS expressing cells (Figure 4F). Altogether, hydroxyurea reduced the frequency of constitutive eNOS and nNOS in bone marrow cells, while NO metabolites downregulated all NOS isoforms.

### 3.5. Level of NOS Isoforms in Mature Erythroid Progenitors after Hydroxyurea and NO Metabolites Treatment

NOS levels were further analyzed in mature CFU-E colonies derived from mouse bone marrow cells previously treated with hydroxyurea or sodium nitrite/nitrate for seven days. Hydroxyurea non-significantly reduced eNOS expressing cells in CFU-E colonies (Figure 5A). Furthermore, sodium nitrite/nitrate reduced the frequency of immunoreactive eNOS cells, and these effects were reversed by PTIO (Figure 5B). Hydroxyurea decreased the frequency of iNOS positive cells in CFU-E, which was reversed by PTIO (*p* < 0.001, Figure 5C). Similar results were obtained with nitrite/nitrate reduction in iNOS positive cells, effects that were completely reversed by PTIO (*p* < 0.001, Figure 5D). The frequency of nNOS positive cells was not significantly changed by hydroxyurea and sodium nitrite/nitrate in CFU-E colonies (not shown). Hydroxyurea demonstrated a NO-dependent reduction in iNOS immunoreactivity in differentiated CFU-E colonies (increased NO levels consequently decreased iNOS expressing cells).

### 3.6. Level of NOS Isoforms in Early Myeloid Progenitors after Hydroxyurea and Nitrite/Nitrate Treatment

Similar to mature CFU-E colonies, we performed NOS isoform analyses on immature BFU-E and CFU-GM colonies harvested from methylcellulose culture. Hydroxyurea and NO metabolites decreased the frequency of eNOS positive cells in BFU-E and CFU-GM colonies (*p* < 0.001), which was not influenced by PTIO (Figure 6A,B). Hydroxyurea did not significantly reduce the frequency of nNOS expressing cells, which was significantly upregulated by PTIO in combined BFU-E and CFU-GM colonies (*p* < 0.01, Figure 6C). The frequency of immunoreactive iNOS cells was not significantly changed by hydroxyurea and sodium nitrite/nitrate in BFU-E and CFU-GM colonies (not shown). Hydroxyurea demonstrated a NO-dependent reduction in nNOS immunoreactivity in differentiated BFU-E and CFU-GM colonies.

## 4. Discussion

Hydroxyurea’s reduction in CFU-E growth and its iNOS frequency have been prevented by NO scavenging (Figure 1). Nitrate reduction in BFU-E growth as well as eNOS and iNOS frequency in CFU-E, later parallel to nitrite, have been reversed by NO scavenging (Figure 1).

We showed NO metabolites reduced the number of bone marrow cells and BFU-E colonies, as well as the NOS frequencies in CFU-E, reversed by NO chelator PTIO. Administration of much stronger ribonucleotide reductase inhibitors than hydroxyurea has been less effective in reducing the number of mouse bone marrow-derived BFU-E and CFU-GM [16]. Therefore, inhibition of ribonucleotide reductase is not principally responsible for hydroxyurea’s reduction in erythroid cell growth. It has been reported that different enzymes (hemoglobin, myoglobin, xanthine oxidoreductase, mitochondrial cytochrome oxidase, cytochrome P450, and others) can catalyze the reduction in nitrite or nitrate to generate NO [17]. The amount, time course, and location (blood vs. tissue) of nitrite/nitrate reduction *in vivo* is unknown, and therefore we evaluated the NO metabolites separately as stable degradation products of NO and hydroxyurea as a NO-releasing agent. Regarding NO-dependence, it is interesting to notice that nitrate reduced the growth of more primitive myeloid progenitors (BFU-E and CFU-GM), while hydroxyurea reduced the growth of mature erythroid progenitors (CFU-E). The reason for this shift can be increasing hemoglobin levels during erythroid differentiation, that react rapidly with NO while sparing hydroxyurea’s inhibition of ribonucleotide reductase [18,19].

Nitrite and nitrate also demonstrated the NO-dependent inhibition of bone marrow cellularity in the presented results. As a NO-releasing agent, hydroxyurea augmented plasma nitrite in humanized sickle cell mice [20]. Moreover, erythrocyte nitrite content was increased, while NOS phosphorylation was decreased in sickle cell anemia patients treated with hydroxyurea [21]. The accumulation of NO metabolites, representing a reversible stable physiological NO reservoir, can simultaneously participate in prolonged myelosuppression during hydroxyurea administration.

We proposed that hydroxyurea inhibits the frequency of constitutive isoforms of NOS in bone marrow cells, as well as the frequency of iNOS expressing cells in CFU-E colonies reversed by NO scavenging. Following this NO dependence, a previous study showed that the NO of nitrosylated hemoglobin derived from hydroxyurea, supporting the NO donor or direct interaction properties of hydroxyurea [22]. NO metabolites also inhibited the frequency of eNOS and iNOS immunoreactive cells in CFU-E colonies, reversed by NO scavenging. It has been reported that PTIO reacts with NO to form imino nitroxides (PTIs) and nitrite, while this nitrite could be reduced again into NO until all PTIO is consumed [23]. Recent reports demonstrated that sickle cell anemia patients have elevated blood nitrite levels which are not further increased by hydroxyurea treatment [24,25]. This is contrary to previous reports that hydroxyurea increased nitrite and nitrate levels with l-arginine [12,26] or without l-arginine in blood [11,21] or erythroid cells [27]. These contrary observations are possibly consistent with whether the patient’s blood is saturated with nitrite or whether it is still sensible to hydroxyurea NO-mediated stimulation of fetal hemoglobin, as well as to a reduction in inflammation and vaso-occlusive crisis.

## 5. Conclusions

This study revealed NO-dependent inhibition of the mature erythroid progenitor (CFU-E) growth by hydroxyurea, demonstrating NO release as a mechanism of action by hydroxyurea. Furthermore, hydroxyurea and NO metabolites downregulated NOS expression in bone marrow cells and myeloid progenitors, while NO metabolites demonstrated NO dependence in the inhibition of bone marrow cellularity. However, the NO scavenger can also collect NO produced from induced NOS, in addition to NO potentially released from hydroxyurea. Therefore, the previously observed NOS activity induced by hydroxyurea [5] cannot be neglected as a mechanism of action of hydroxyurea in erythroid progenitors, but this study generally supports NO dependence. Moreover, hydroxyurea’s reduction in the iNOS level was also NO-dependent, demonstrating a negative feedback mechanism in the control of NO production via NOS in mature erythroid progenitors. The NO-releasing properties of hydroxyurea importantly contribute to the molecular basis of its pharmacological features.

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
