# Peer review of "Nitric Oxide Mediation in Hydroxyurea and Nitric Oxide Metabolites’ Inhibition of Erythroid Progenitor Growth"

_biomolecules, 2021, doi:10.3390/biom11111562_

Round 1
Reviewer 1 Report
Comments to the authors
Overall comment:
(1) The authors should be congratulated on having large number of replications; N = 5 (Fig 5), n = 6 (Fig 1); n=10 (Figs 2, 3, 4).
(2) The authors repeat the chemical name for PTIO (2-phenyl-4,4,5,5-tetramethylimidazoline-1-oxyl-3-oxide) multiple times throughout the article. Please state it once and use the abbreviation afterwards.
(3) on Figs 1B-C, 2A-B, 3A-F, (etc) please indicate which bars refer to vehicle-injected mice. The authors provide this information on Fig 1A but not in any successive figure.
(4) The immunofluorescent panels (Figs 4-6) are extremely small and very difficult to evaluate. Would it be possible to include larger panels. (maybe include larger versions as supplemental information?)
(5) The author might consider look at the entire data set to infer conclusions from the data. For example, on Fig 3A, PTIO reversed hydroxyurea (HU) inhibition of BFU-E colonies and the authors conclude that HU acts via nitric oxide (NO). On Fig 3B, nitrite and nitrate also inhibited BFU-E colonies, an effect not reversed by PTIO. One might conclude that nitrite and nitrate did not act via NO. The authors assume in the paper is that nitrite and nitrate are reduced into NO by physiological processes (see for example line 164 where they are referred as “NO-releasing agents”).
A similar situation occurs in Fig 4 (A-F). HU, nitrite and nitrate reduced expression of eNOS and nNOS, but not iNOS. The authors conclude that eNOS and nNOS expression are controlled by NO. But PTIO did not reverse any of these effects. Therefore, how can such effects be mediated by NO?
The reader is led to believe by the authors that if PTIO reversed an effect mediated by HU, then HU was acting via NO (ex: Fig 3A). However, if an effect was mediated by nitrite and nitrate and not reversed by PTIO, then nitrite and nitrate were not acting via NO (ex: Fig 3D). But nitrite/nitrate are assumed to be NO-donors. At the same time, if an effect was induced by HU and not reversed by PTIO but was similar to the effects induced by nitrite and nitrate, then HU was also acting via NO (ex: Fig 4A-F). To compound all of this logical confusion PTIO (by itself, without any co-treatment) produced results in a number of assays (ex: Fig 5 A-C).
The only instance where results for HU, nitrite and nitrate are in accord and, also, are all reversed by PTIO is for Fig 5C-D (iNOS expression in CFU-E colonies).
(6) The author should acknowledge that it is unknown how much of the administered nitrite and nitrate is converted into NO in vivo. The bioactivation of nitrite by enzymes (hemoglobin, xanthine oxido-reductase and others) and non-enzymatic processes (ascorbic acid in acidic stomach environment) has been intensely studied (Amdahl et al, Biol. Chem. 401(1): 201–211, 2020). But it is unknown the amount, time course and location (blood vs tissue) of nitrite reduction in vivo. Plus, nitrite reduction is proposed to occur preferentially at sites of hypoxia/acidity (ex: ischemic tissue), which should not be present in a common mouse strain under routine care. Therefore, to use nitrite and nitrate as in vivo surrogates for NO-donors and to expect that its pharmacokinetics will reproduce NO released by HU is unrealistic.
The use of PTIO as a NO chelator in vivo should be carefully interpreted. PTIO reacts with NO to form PTI and nitrite and it is conceivable that this nitrite could be reduced again into NO, until all PTIO is consumed (see Hogg et al, Free Rad. Res., 22(1): 41-56, 1995 for the original research into PTIO and NO reaction and Goldstein et al, JBC 278(51): 50949–50955, 2003 for a more recent kinetic analysis). In this way, PTIO may actually boosts NO signaling by chelating it and re-forming its precursor (nitrite). However, PTIO (usually its derivative c-PTIO) is largely used in the literature to indicate NO-mediated effects (ex: Hirano et al, OncoImmunology 4:8, e1019195, 2015; Peñarando et al, BMC Biology 16:3, 2018; Lee et al, Asian Journal of Andrology 21: 92–97, 2019).
Given the above considerations, the authors might consider restrain from overinterpreting data which includes HU +/- PTIO, without trying to compare it with nitrite/nitrate effects. For example, under such analysis, HU-mediated and PTIO-sensitive effects: Fig 3C and 5C. I believe that these effects can be ascertained as HU-induced and NO-mediated.
On the other hand, several effects induced by HU were PTIO-insensitive: Fig 2A, 3A, 3E, 4A, 4C, 5A and 6A. These effects do not appear to involve NO release from HU.
The data for nitrite and nitrate should be included (as shown by the authors) but their relationship to the HU data, if any, is for the reader to evaluate.
Other comments
Line 34:
“Plasma levels of nitrite and nitrate are increased in patients with sickle cell disease and essential thrombocythemia on hydroxyurea therapy [1-10]”
This statement as it relates to sickle cell disease (SCD) is disputed in the literature. Reference #1 cited by authors (Gladwin et al, Br. J. Haematol. 116:436-444, 2002) indicated that plasma nitrite, nitrate (and nitrosyl-Hb) levels are elevated in SCD patients after HU administration, but that study did not compare SCD patients on or off HU treatment. Others (Almeida et al, Nitric Oxide, 94: 79–91, 2020) indicated that all SCD patients have elevated blood nitrite levels, independently of HU treatment. However, there are several studies indicating that HU may act in SCD therapy via NO production. For example, please see Jiang et al (Mol Pharm, 52:1081-1086, 1997) and Glover et al (Mol Pharm 55:1006-1010, 1999). Perhaps the authors could clarify that it is not established that HU increases blood nitrite/nitrate levels in all SCD patients.
Line 37:
Please define “NOx”
Line 37:
“(…) production of NOx by hydroxyurea is supported by the NOS substrate l-arginine in erythroid cells [12]”
Reference #12 (Morris et al, J. Pediatr. Hematol. Oncol. 25(8): 629-634, 2003) administered l-arginine with/without hydroxyurea to SCD patients and assayed plasma NOx levels, but did not demonstrated that l-arginine acted on erythroid cells.
Section “colony forming assays” (lines 51-71):
Authors refer to previous publication for methodological details (ref #5), but some additional details should be included in this work to help the reader.
Please add the mouse strain used in the study
Please add the vehicle used for tail vein injection of hydroxyurea, PTIO, nitrite and nitrate (ex: PBS, saline, etc).
Please add the volume injected on each mouse (example: “10 µl/g”, etc).
Please, state if solutions (PTIO, hydroxyurea, nitrite or nitrate) were sterilized with syringe filters before injection (if yes, please give catalog # and manufacturer). Some of these filters have large amounts of contaminating nitrite and nitrate and could interfere with results.
Please indicate if nitrite and nitrate were assayed in the culture media (methylcellulose media, with supplements). Some (but not all) cell culture mediums can contain nitrite and nitrate but these substances are not assayed and reported by the manufacturer. The presence (if any) of nitrite and/or nitrate in culture media could have influenced results.
Section “hematologic parameters” (lines 72-81):
Please briefly state how mouse blood was collected (ex: “retro-orbital sinus”, “cardiac puncture after CO2 euthanasia”, etc),
Line 78:
“Briefly, 100µM of each sample and each standard were pipetted (…)”
Did the authors mean “100µl”. Please correct if appropriate.
Section “immunocytochemistry” (lines 82-92):
Please give the catalog number (cat#) for each antibody and their respective dilutions. Antibodies for immunostaining are notoriously variable within a single manufacturer and this information can help the reader.
Table 1:
Please state the units of measurements for hematocrit and reticulocytes.
Please state the number of animals used for this experiment .
Line 169:
The word “quantity” should be replaced by “frequency”, as all effects are presented as relative (as %) and not as absolute changes.
Line 227:
“(…) PTIO prevented hydroxyurea’s reduction of reticulocytes and thrombocytopenia”
This sentence is true only for reticulocytes, but not for thrombocytes. Table 1 shows that HU and PTIO induced thrombocytopenia (***, p<0.001 for each drug vs control) and the combination (HU+PTIO, *** p<0.001 vs control) did not reverse the effects induced by HU.
Author Response
REVIEWER 1
Overall comment:
(1) The authors should be congratulated on having large number of replications; N = 5 (Fig 5), n = 6 (Fig 1); n=10 (Figs 2, 3, 4).
Thank you.
(2) The authors repeat the chemical name for PTIO (2-phenyl-4,4,5,5-tetramethylimidazoline-1-oxyl-3-oxide) multiple times throughout the article. Please state it once and use the abbreviation afterwards.
We used the chemical name for PTIO in Abstract, Materials and Methods and Figure legends. We will keep the chemical name for PTIO Abstract and Materials and Methods, while we will use abbreviation in Figure legends.
(3) on Figs 1B-C, 2A-B, 3A-F, (etc) please indicate which bars refer to vehicle-injected mice. The authors provide this information on Fig 1A but not in any successive figure.
We added the Control samples in all Figures and their legends.
(4) The immunofluorescent panels (Figs 4-6) are extremely small and very difficult to evaluate. Would it be possible to include larger panels. (maybe include larger versions as supplemental information?)
The immunofluorescent panels (Figs 4-6) are presented as larger versions in corresponding Supplemental Figures 1 – 3.
(5) The author might consider look at the entire data set to infer conclusions from the data. For example, on Fig 3A, PTIO reversed hydroxyurea (HU) inhibition of BFU-E colonies and the authors conclude that HU acts via nitric oxide (NO). On Fig 3B, nitrite and nitrate also inhibited BFU-E colonies, an effect not reversed by PTIO. One might conclude that nitrite and nitrate did not act via NO. The authors assume in the paper is that nitrite and nitrate are reduced into NO by physiological processes (see for example line 164 where they are referred as “NO-releasing agents”).
Thank you for your detailed comment. We suppose that you are addressing Figs 3C,D instead of Figs 3A,B. On Fig 3A PTIO failed to reverse hydroxyurea (HU) inhibition of BFU-E. However in Fig 3B, PTIO significantly reversed nitrate inhibition of BFU-E colonies, while PTIO diminished significance of nitrite inhibition of BFU-E colonies. It is true in Fig 3C that PTIO reversed HU inhibition of CFU-E, but failed to reverse nitrite and nitrate inhibition of CFU-E colonies in Fig 3D. So, the PTIO effects on HU and nitrite/nitrate do not follow each other in applied concentrations. In the final sentence of this Results section, to distinguish HU and nitrite/nitrate effects, we made the following statement: “Hydroxyurea and NO metabolites nitrite and nitrate generally inhibited the myeloid colony growth indicating NO dependence for hydroxyurea in the CFU-E colonies and for nitrate (NO3) in the BFU-E and CFU-GM colonies in applied concentrations.”
Regarding previous reports, it is shown that nitrite and nitrate represent the final products of nitric oxide (NO) oxidation pathways (1). Indeed, nitrite and nitrate are the stable degradation products of NO that accumulate in supernatants of biological samples. Under certain conditions, different enzymes (hemoglobin, myoglobin, xanthineoxidoreductase, mitochondrial cytochrome oxidase, aldehyde de-hydrogenase 2, cytochrome P450 reductase and cytochrome P450) can catalyze the reduction of nitrite or nitrate to generate NO (2). Sickle cell anaemia patients taking hydroxyurea (HU) show a significant increase in plasma nitrite and nitrate within 2 h of ingestion, providing evidence for the in vivo conversion of HU to NO (3). Supplementation with L-arginine plus HU demonstrated a significant increase in nitrite and nitrate levels in sickle cell anaemia (4). Moreover, HU therapy is an independent predictor of higher nitrite and nitrate levels in ET (5). References:
- Giustarini D, Rossi R, Milzani A, Dalle-Donne I. Nitrite and nitrate measurement by Griess reagent in human plasma: evaluation of interferences and standardization. Methods Enzymol. 2008;440:361-80.
- Zhao Y, Vanhoutte PM, Leung SW. Vascular nitric oxide: Beyond eNOS. J Pharmacol Sci. 2015;129(2):83-94.
- Gladwin MT, Shelhamer JH, Ognibene FP, Pease-Fye ME, Nichols JS, Link B, Patel DB, Jankowski MA, Pannell LK, Schechter AN, Rodgers GP. Nitric oxide donor properties of hydroxyurea in patients with sickle cell disease. Br J Haematol. 2002;116(2):436-44.
- Eleutério RMN, Nascimento FO, Araújo TG, Castro MF, Filho TPA, Filho PAM, Eleutério J Jr, Elias DBD, Lemes RPG. Double-Blind Clinical Trial of Arginine Supplementation in the Treatment of Adult Patients with Sickle Cell Anaemia. Adv Hematol. 2019;2019:4397150.
- Cella G, Marchetti M, Vianello F, Panova-Noeva M, Vignoli A, Russo L, Barbui T, Falanga A. Nitric oxide derivatives and soluble plasma selectins in patients with myeloproliferative neoplasms. Thromb Haemost. 2010;104(1):151-6.
In this study, we wanted to observe a difference between HU effects and its stable degradation products of NO production: nitrite and nitrate. We did not observe augmented NO levels in plasma of mouse peripheral blood after individual treatment with HU, nitrite and nitrate (Figure 1). Considering previously reported enzymatic mediation of nitrite/nitrate reduction to NO in blood (2), we assumed the nitrite and nitrate as potentially “NO-releasing agents” in blood. However, because we did not demonstrate their induction of NO levels, we will retreat to mention them as “NO-releasing agents” in manuscript.
A similar situation occurs in Fig 4 (A-F). HU, nitrite and nitrate reduced expression of eNOS and nNOS, but not iNOS. The authors conclude that eNOS and nNOS expression are controlled by NO. But PTIO did not reverse any of these effects. Therefore, how can such effects be mediated by NO?
Thank you for comments regarding Fig 4. I apologize for misunderstanding, but we did not mention that “eNOS and nNOS expression are controlled by NO”. In Results section for Fig 4 we mentioned: “Taken together, hydroxyurea reduced the frequency of constitutive eNOS and nNOS, similar to NO metabolites, but did not downregulate iNOS positive cells.” To clarify this, we changed the last sentence into: “Altogether, hydroxyurea reduced the frequency of constitutive eNOS and nNOS in bone marrow cells, while NO metabolites downregulated all NOS isoforms.”
The reader is led to believe by the authors that if PTIO reversed an effect mediated by HU, then HU was acting via NO (ex: Fig 3A). However, if an effect was mediated by nitrite and nitrate and not reversed by PTIO, then nitrite and nitrate were not acting via NO (ex: Fig 3D). But nitrite/nitrate are assumed to be NO-donors. At the same time, if an effect was induced by HU and not reversed by PTIO but was similar to the effects induced by nitrite and nitrate, then HU was also acting via NO (ex: Fig 4A-F). To compound all of this logical confusion PTIO (by itself, without any co-treatment) produced results in a number of assays (ex: Fig 5 A-C). The only instance where results for HU, nitrite and nitrate are in accord and, also, are all reversed by PTIO is for Fig 5C-D (iNOS expression in CFU-E colonies).
Thank you for very detailed comments. In the Conclusion we had statement: “Therefore, the previously observed NOS activity induced by hydroxyurea [5], cannot be neglected as a mechanism of action of hydroxyurea in erythroid progenitors, but this study supports NO dependence.” I suppose that you are mixing Fig 3A and 3C, so you are addressing Fig 3C that PTIO reversed an effect mediated by HU to myeloid progenitors. As NO scavenger reversed HU inhibition of CFU-E colonies, we can not neglect NO dependence in HU activity. Nitrite and nitrate as stable degradation products of NO can be reduced again to NO in blood to mimic antiproliferative NO-releasing agent such as HU. In standardized concentrations of applied agents, we did not observe steady effects of PTIO on ex vivo myeloid progenitor growth stimulated either by HU or NO metabolites (including Fig 3D). HU and nitrite/nitrate effects on NOS frequency in bone marrow cells (Fig 4) were not addressed as linked in Results section. We just observed HU reduction of constitutive NOS frequency not affected by PTIO in bone marrow cells. Generally, NO metabolites and HU as NO-releasing agents reduced NOS frequency by negative feedback but applied concentrations of PTIO failed to reverse effects. In contrast we can tell that NO metabolites and HU do not use NO to control NOS frequency because PTIO failed to reverse effects in bone marrow cells. This can be supported by observation that PTIO, as an NO scavenger, by itself reduced eNOS and iNOS frequency both in mature (Fig 5) and immature (Fig 6) myeloid progenitors. However, PTIO reversed nitrite/nitrate effects on eNOS/iNOS as well as HU on iNOS in mature myeloid progenitors (Fig 5). Considering previous, we observed NO (PTIO) dependency of HU preferentially in CFU-E colonies during differentiation. NO (PTIO) dependencies of nitrite/nitrate were individually or commonly presented in bone marrow cells and myeloid progenitors. These were described in Discussion and Conclusion.
(6) The author should acknowledge that it is unknown how much of the administered nitrite and nitrate is converted into NO in vivo. The bioactivation of nitrite by enzymes (hemoglobin, xanthine oxido-reductase and others) and non-enzymatic processes (ascorbic acid in acidic stomach environment) has been intensely studied (Amdahl et al, Biol. Chem. 401(1): 201–211, 2020). But it is unknown the amount, time course and location (blood vs tissue) of nitrite reduction in vivo. Plus, nitrite reduction is proposed to occur preferentially at sites of hypoxia/acidity (ex: ischemic tissue), which should not be present in a common mouse strain under routine care. Therefore, to use nitrite and nitrate as in vivo surrogates for NO-donors and to expect that its pharmacokinetics will reproduce NO released by HU is unrealistic.
Thank you for describing the nitrite and nitrate conversion into NO. We added in Abstract:”…we individually examined hydroxyurea (NO releasing agent) and NO metabolites (stable NO degradation products) effects on…” We added also in Discussion section, second paragraph: “It has been reported that different enzymes (hemoglobin, myoglobin, xanthine oxidoreductase, mitochondrial cytochrome oxidase, cytochrome P450 and others) can catalyze the reduction of nitrite or nitrate to generate NO [17]. It is unknown the amount, time course and location (blood vs. tissue) of nitrite/nitrate reduction in vivo and therefore we evaluated separately the NO metabolites as stable degradation products of NO and hydroxyurea as a NO releasing agent.”
- Amdahl MB, DeMartino AW, Gladwin MT. Inorganic nitrite bioactivation and role in physiological signaling and therapeutics. Biol Chem. 2019 Dec 18;401(1):201-211.
The use of PTIO as a NO chelator in vivo should be carefully interpreted. PTIO reacts with NO to form PTI and nitrite and it is conceivable that this nitrite could be reduced again into NO, until all PTIO is consumed (see Hogg et al, Free Rad. Res., 22(1): 41-56, 1995 for the original research into PTIO and NO reaction and Goldstein et al, JBC 278(51): 50949–50955, 2003 for a more recent kinetic analysis). In this way, PTIO may actually boosts NO signaling by chelating it and re-forming its precursor (nitrite). However, PTIO (usually its derivative c-PTIO) is largely used in the literature to indicate NO-mediated effects (ex: Hirano et al, OncoImmunology 4:8, e1019195, 2015; Peñarando et al, BMC Biology 16:3, 2018; Lee et al, Asian Journal of Andrology 21: 92–97, 2019).
Thank you for detailed explanation and references for PTIO. We added in last paragraph of Discussion: “It has been reported that PTIO reacts with NO to form imino nitroxides (PTIs) and nitrite, while this nitrite could be reduced again into NO, until all PTIO is consumed [23].”
- Goldstein, S.; Russo, A.; Samuni, A. Reactions of PTIO and carboxy-PTIO with *NO, *NO2, and O2-*. J Biol Chem. 2003, 278, 50949-50955.
Given the above considerations, the authors might consider restrain from overinterpreting data which includes HU +/- PTIO, without trying to compare it with nitrite/nitrate effects. For example, under such analysis, HU-mediated and PTIO-sensitive effects: Fig 3C and 5C. I believe that these effects can be ascertained as HU-induced and NO-mediated. On the other hand, several effects induced by HU were PTIO-insensitive: Fig 2A, 3A, 3E, 4A, 4C, 5A and 6A. These effects do not appear to involve NO release from HU.
In accordance with this and previous comments, we consistently avoided to compare HU and nitrite/nitrate effects throughout the manuscript. We insisted in Results and Discussion sections on HU-mediated and PTIO-sensitive effects in Fig 3C and 5C.
The data for nitrite and nitrate should be included (as shown by the authors) but their relationship to the HU data, if any, is for the reader to evaluate.
The data for nitrite and nitrate are presented separately and not observed in respect to HU results.
Other comments
Line 34:
“Plasma levels of nitrite and nitrate are increased in patients with sickle cell disease and essential thrombocythemia on hydroxyurea therapy [1, 10]”
This statement as it relates to sickle cell disease (SCD) is disputed in the literature. Reference #1 cited by authors (Gladwin et al, Br. J. Haematol. 116:436-444, 2002) indicated that plasma nitrite, nitrate (and nitrosyl-Hb) levels are elevated in SCD patients after HU administration, but that study did not compare SCD patients on or off HU treatment. Others (Almeida et al, Nitric Oxide, 94: 79–91, 2020) indicated that all SCD patients have elevated blood nitrite levels, independently of HU treatment. However, there are several studies indicating that HU may act in SCD therapy via NO production. For example, please see Jiang et al (Mol Pharm, 52:1081-1086, 1997) and Glover et al (Mol Pharm 55:1006-1010, 1999). Perhaps the authors could clarify that it is not established that HU increases blood nitrite/nitrate levels in all SCD patients.
Regarding reference by Almeida et al, 2012, PTIO reversed the beneficial properties of HU with respect to leukocyte adhesion and extravasation in inflammatory mice, while treatment with L-NAME did not have significant effects. They suggested that HU acts as a NO donor rather than via stimulation of NOS activity in inflammation. Regarding reference by Almeida et al, 2020, SCD in humans and animals is associated with increased nitrite/NO availability, while in vivo inhibition of NO synthases decreased nitrite levels in homozygous SCD mice, where injection of a NO scavenger c-PTIO decreased bleeding time. HU did not significantly increase nitrite levels in whole blood, red blood cells and plasma of SCD patients. In addition, Grau et al, 2015, showed that plasma nitrite content was not different between SCD patients and healthy controls, while RBC-NOS activation is higher in SCD.
Several studies demonstrated that HU increased NO metabolites level in:
- L-arginine plus HU increased nitrite and nitrate levels in SCD patients after 4 months. [26]
- the plasma levels of the NO derivatives (NOX) (nitrites and nitrates) were significantly increased in ET patients treated with HU [11].
- HU treatment increased nitrate and nitrite levels in K562 and primary erythroid cells [27]
- RBC nitrite content was higher in SCD under HU treatment than in SCD without HU treatment and healthy subjects [21]
In accordance to reviewer suggestion to clarify that it is not established that HU increases blood nitrite/nitrate levels in all SCD patients, we added the following statement at the end of Discussion: “Recent reports demonstrated sickle cell anemia patients have elevated blood nitrite levels, not further increased by hydroxyurea treatment [24,25]. It is opposite to previous reports that hydroxyurea increased nitrite and nitrate levels with l-arginine [12,26] or without l-arginine in blood [11,21] or erythroid cells [27]. These opposite observations are possibly in accordance to patient blood saturated with nitrite or still sensible for NO-mediated hydroxyurea stimulation of fetal hemoglobin as well as reduction of inflammation and vaso-occlusive crises.”
Line 37:
Please define “NOx”
NOx is total NO level (NOx). We included full name in manuscript.
Line 37:
“(…) production of NOx by hydroxyurea is supported by the NOS substrate l-arginine in erythroid cells [12]”
Reference #12 (Morris et al, J. Pediatr. Hematol. Oncol. 25(8): 629-634, 2003) administered l-arginine with/without hydroxyurea to SCD patients and assayed plasma NOx levels, but did not demonstrated that l-arginine acted on erythroid cells.
Thank you for correction, we mixed it with a new reference 28. We corrected now in: “Production of total NO level (NOx) by hydroxyurea is supported by the NOS substrate l-arginine in blood [12].”
Section “colony forming assays” (lines 51-71):
Authors refer to previous publication for methodological details (ref #5), but some additional details should be included in this work to help the reader.
Please add the mouse strain used in the study.
We included in Materials and Methods (section Colony forming assays): “..the male CBA mice (6-8 weeks old) were treated daily...””
Please add the vehicle used for tail vein injection of hydroxyurea, PTIO, nitrite and nitrate (ex: PBS, saline, etc).
We included in Materials and Methods (section Colony forming assays): “…with NO scavenger 2-phenyl-4,4,5,5-tetramethylimidazoline-1-oxyl 3-oxide (PTIO, 1 mg/kg in pro injectione H2O, Santa Cruz Biotechnologies, Dallas, Texas, USA) and hydroxyurea (200 mg/kg in phosphate-buffered saline (PBS), Sigma-Aldrich, St. Louis, Missouri, USA) with or without preincubation of 30 minutes with PTIO (1 mg/kg, Sig-ma-Aldrich) for 7 consecutive days. In addition, animals were treated with sodium nitrite (1 mmol/kg in PBS, Sigma-Aldrich) or sodium nitrate (1 mmol/kg in PBS, Sigma-Aldrich) with or...”
Please add the volume injected on each mouse (example: “10 µl/g”, etc).
We included in Materials and Methods (section Colony forming assays): “The total volume injected on each mouse was 10 µl/g (for one shot, per 5 µl/g for two shots).“
Please, state if solutions (PTIO, hydroxyurea, nitrite or nitrate) were sterilized with syringe filters before injection (if yes, please give catalog # and manufacturer). Some of these filters have large amounts of contaminating nitrite and nitrate and could interfere with results.
The solutions (PTIO, hydroxyurea, nitrite or nitrate) were not sterilized with syringe filters before injection.
Please indicate if nitrite and nitrate were assayed in the culture media (methylcellulose media, with supplements). Some (but not all) cell culture mediums can contain nitrite and nitrate but these substances are not assayed and reported by the manufacturer. The presence (if any) of nitrite and/or nitrate in culture media could have influenced results.
The nitrite and nitrate were not assayed in the culture methylcellulose media. All mouse bone marrow cells (control and treated) were cultured in the same methylcellulose media. Nitrite and nitrate are not reported by the manufacturer. MethoCult™ M3334 contains:
- Methylcellulose in Iscove's MDM
- Fetal bovine serum
- Bovine serum albumin
- Recombinant human insulin
- Human transferrin (iron-saturated)
- 2-Mercaptoethanol
- Recombinant human erythropoietin (EPO)
- Supplements
MethoCult™ GF M3434 in addition contains:
- Recombinant mouse stem cell factor (SCF)
- Recombinant mouse interleukin 3 (IL-3)
- Recombinant human interleukin 6 (IL-6)
Section “hematologic parameters” (lines 72-81):
Please briefly state how mouse blood was collected (ex: “retro-orbital sinus”, “cardiac puncture after CO2 euthanasia”, etc),
We included in Materials and Methods (section Hematologic parameters): “Mouse blood was collected from retro-orbital sinus.”
Line 78:
“Briefly, 100µM of each sample and each standard were pipetted (…)”
Did the authors mean “100µl”. Please correct if appropriate.
Thank you, we corrected to “100 µl”.
Section “immunocytochemistry” (lines 82-92):
Please give the catalog number (cat#) for each antibody and their respective dilutions. Antibodies for immunostaining are notoriously variable within a single manufacturer and this information can help the reader.
We included in Materials and Methods (section Immunocytochemistry): “Briefly, after fixation and incubation with the anti-neuronal NOS (nNOS, cat no. sc-5302 (1:100), Santa Cruz Biotechnology), anti-inducible NOS (iNOS, cat. no. sc-651 (1:100), San-ta Cruz Biotechnology), and anti-eNOS antibodies (cat. no. sc-654 (1:100), Santa Cruz Bio-technology) immunostaining…”
Table 1:
Please state the units of measurements for hematocrit and reticulocytes.
We included the units of measurements for hematocrit (l/l) and reticulocytes (%) in Table 1. In addition, we clarified % of reticulocytes in Table legend: “1reticulocytes per 1000 erythrocytes.”
Please state the number of animals used for this experiment.
We state the number of mice in column Treatment (n=7), with description in Table legend: “n – number of studied mice per group;”
Line 169:
The word “quantity” should be replaced by “frequency”, as all effects are presented as relative (as %) and not as absolute changes.
We replaced the word “quantity” by “frequency” throughout the manuscript where the effects are presented as relative (as %).
Line 227:
“(…) PTIO prevented hydroxyurea’s reduction of reticulocytes and thrombocytopenia”
This sentence is true only for reticulocytes, but not for thrombocytes. Table 1 shows that HU and PTIO induced thrombocytopenia (***, p<0.001 for each drug vs control) and the combination (HU+PTIO, *** p<0.001 vs control) did not reverse the effects induced by HU.
Thank you for comment. PTIO just partially prevented hydroxyurea’s reduction of thrombocytes, but not significantly, so we will remove “PTIO prevented hydroxyurea’s reduction of thrombocytopenia”.
Thank you very much for detailed and constructive review of the manuscript. It changed the concept, clarify, and increase the quality of presented results. We appreciated your support.

Reviewer 2 Report
This paper is of interest as it explains the effects of nitric oxide on erythroid progenitors cells.
However, the major failure of the paper is the way it is written:
INTRODUCTION:
In its current format it is not easy to read. Authors should simplify it.
Ideally, the text should be made smooth and by steps.
Concept should be explained.
Where possible a schematic picture of the pathophysiology of nitric oxide and eNOs should be added
Also, I believe that it will benefit if it could be revised by an English mother tongue speaker.
I would like to recommend to look at these papers and to use them as references:
Eur J Haematol. 2019 Apr;102(4):319-330. doi: 10.1111/ejh.13212
J Ir Dent Assoc. 2008 Apr-May;54(2):75-
Ann Hematol. 2012 Oct;91(10):1669-71. doi: 10.1007/s00277-012-1447-9
METHODS: Ok
RESULTS: OK
DISCUSSION:
This part should be fully rewritten.
It looks like a list of additional results.
This should be moved within RESULTS section.
Please note: within discussion authors needs to be critical on the results and should add their comment
This should take into account critical about pros and cons of their finding and refer to previously written articles with similar and different finding.
Author Response
REVIEWER 2
This paper is of interest as it explains the effects of nitric oxide on erythroid progenitors cells.
However, the major failure of the paper is the way it is written:
INTRODUCTION:
In its current format it is not easy to read. Authors should simplify it. Ideally, the text should be made smooth and by steps.
Thank you for comments, we rearranged and simplify the Introduction to make it easier to read. We presented hydroxyurea as NO releasing agent through preclinical in vitro and animal studies in first paragraph, while clinical studies of hydroxyurea as NO releasing agent are presented in second paragraph. NO metabolite studies and their link with hydroxyurea are presented in third paragraph. This smoothly step by step approach is completed with a concept and short description of the study in the final paragraph of Introduction.
Concept should be explained.
The concept of the study is explained in the first sentence of the last paragraph in Introduction: ”We now question whether hydroxyurea primarily acts as an NO donor or cumulatively increase the reversible NO metabolites participating in myelosuppression.” In the last sentence of the third paragraph we additionally supported the concept of the study: ”As the final products of NO oxidation pathways and reversible NO metabolites [29], nitrite and nitrate may have the potential to distract or overlap hydroxyurea efficacy as a NO releasing agent.” The concept of experimental performance is presented in second sentence of the last paragraph: ”To observe this, we use mice for in vivo parallel studies with hydroxyurea and final products of NO (nitrite and nitrate) in co-treatment with a NO chelator PTIO that does not influence the NOS activity.”
Where possible a schematic picture of the pathophysiology of nitric oxide and eNOS should be added. Also, I believe that it will benefit if it could be revised by an English mother tongue speaker.
Thank you for advice for a schematic picture, we included in Discussion section a Scheme I to clearly present the results of the study, with a legend: “Scheme I. Nitric oxide (NO) dependent (by PTIO) hydroxyurea (HU), nitrite (NO2) and nitrate (NO3) effects on myeloid differentiation and frequency of NOS isoforms in myeloid progenitors.” In accordance with the reviewer statement that “English language and style are fine/minor spell check required” and further major comment, we additionally checked the spelling and grammar throughout the manuscript.
I would like to recommend to look at these papers and to use them as references:
Eur J Haematol. 2019 Apr;102(4):319-330. doi: 10.1111/ejh.13212
J Ir Dent Assoc. 2008 Apr-May;54(2):75-
Ann Hematol. 2012 Oct;91(10):1669-71. doi: 10.1007/s00277-012-1447-9
Thank you for recommendation to read the interesting papers. We found the first one: “Eur J Haematol. 2019 Apr;102(4):319-330.” suitable for reference in Introduction section (reference 2).
METHODS: Ok
Thank you.
RESULTS: OK
Thank you.
DISCUSSION:
This part should be fully rewritten. It looks like a list of additional results. This should be moved within RESULTS section.
The first paragraph with results in Discussion has been removed. In the beginning, we briefly described the Scheme I.
Please note: within discussion authors needs to be critical on the results and should add their comment. This should take into account critical about pros and cons of their finding and refer to previously written articles with similar and different finding.
Thank you for useful comments. After short description of the Scheme I, we rearranged discussion on the following way:
- First paragraph, we described critically our results with NO metabolites and myeloid progenitor growth and correlated with hydroxyurea mechanism of action. We applied also references and made comments regarding parallel myelosuppression.
- Second paragraph, we described our results with NO metabolites and bone marrow cellularity and refer to previously written articles with hydroxyurea induction of nitrite levels. We added our comment in the final sentence.
- Third paragraph, we presented critically our results with NO metabolites and hydroxyurea and their influence on NOS isoforms in erythroid progenitors. We consider critical about pros and cons of NO dependence, for hydroxyurea and nitrite/nitrate, and refer to previously written articles with similar and different results. We added our comments evaluating presented and previously reported data.
- Conclusion, considering presented results we made short overview and conclusions related to hydroxyurea mechanism of action.
Thank you very much for critical review of the manuscript. Your comments and suggestions clarify the presented results and their description.

Round 2
Reviewer 1 Report
The authors were very attentive to our suggestions and responded to all of our comments adequately. We are satisfied with all the clarifications added to the text. The manuscript has greatly improved.
Reviewer 2 Report
accept!